# Epidemiological Analysis of Cassava Mosaic and Brown Streak Diseases, and *Bemisia tabaci* in the Comoros Islands

**DOI:** 10.3390/v14102165

**Published:** 2022-09-30

**Authors:** Rudolph Rufini Shirima, Everlyne Nafula Wosula, Abdou Azali Hamza, Nobataine Ali Mohammed, Hadji Mouigni, Salima Nouhou, Naima Mmadi Mchinda, Gloria Ceasar, Massoud Amour, Emmanuel Njukwe, James Peter Legg

**Affiliations:** 1International Institute of Tropical Agriculture (IITA-Tanzania), P.O. Box 34441, Dar es Salaam 14112, Tanzania; 2Institut National de Recherche pour L’Agriculture, La Pêche et L’Environnement (INRAPE), Moroni BP 1406, Comoros; 3West and Central African Council for Agricultural Research and Development (CORAF), Dakar CP 18523, Senegal

**Keywords:** surveillance, CBSD, CMD, CBSIs, CMBs, *Bemisia tabaci*, HTS

## Abstract

A comprehensive assessment of cassava brown streak disease (CBSD) and cassava mosaic disease (CMD) was carried out in Comoros where cassava yield (5.7 t/ha) is significantly below the African average (8.6 t/ha) largely due to virus diseases. Observations from 66 sites across the Comoros Islands of Mwali, Ngazidja, and Ndzwani revealed that 83.3% of cassava fields had foliar symptoms of CBSD compared with 95.5% for CMD. Molecular diagnostics confirmed the presence of both cassava brown streak ipomoviruses (CBSIs) and cassava mosaic begomoviruses (CMBs). Although real-time RT-PCR only detected the presence of one CBSI species (*Cassava brown streak virus*, CBSV) the second species (*Ugandan cassava brown streak virus*, UCBSV) was identified using next-generation high-throughput sequencing. Both PCR and HTS detected the presence of East African cassava mosaic virus (EACMV). African cassava mosaic virus was not detected in any of the samples. Four whitefly species were identified from a sample of 131 specimens: *Bemisia tabaci*, *B. afer*, *Aleurodicus dispersus*, and *Paraleyrodes bondari*. Cassava *B. tabaci* comprised two mitotypes: SSA1-SG2 (89%) and SSA1-SG3 (11%). KASP SNP genotyping categorized 82% of cassava *B. tabaci* as haplogroup SSA-ESA. This knowledge will provide an important base for developing and deploying effective management strategies for cassava viruses and their vectors.

## 1. Introduction

Cassava (*Manihot esculenta* Crantz) is cultivated in many countries in sub-Saharan Africa (SSA) where it supports the livelihoods of more than 500 million people as a source of food and income [1]. Although Africa hosts the largest proportion of the world’s cassava-cultivated land (>80%, [2]), farmers do not fully benefit from the crop’s yield potential due to the many pests and diseases, and poor agricultural management practices resulting from lack of the resources and knowledge to practice profitable and sustainable farming [3]. The most devastating diseases are cassava mosaic disease (CMD) and cassava brown streak disease (CBSD) which are caused by viruses [4]. CMD is present in all cassava-growing areas in Africa and can account locally for over 80% of cassava root yield losses [5]. The last continent-wide assessment of losses due to CMD provided an estimate of greater than 30 million tonnes annually [6].

CBSD is caused by cassava brown streak ipomoviruses (CBSIs; which is the standard name for the two cassava brown streak viruses together) [7,8,9,10] and CMD by cassava mosaic begomoviruses (CMBs) [11,12,13]. Both groups of viruses are propagated by infected planting material and transmitted by the whitefly vector, *Bemisia tabaci* (Genn.) [14,15]. *B. tabaci* comprises many morphologically identical populations but with genetically distinct mitotypes designated based on the partial sequencing of the *COI* gene [16,17]. The cassava-colonizing group is subdivided into several mitotype sub-groups (SG) [18,19,20,21]. A more recent study used more than 60,000 SNPs to reassign all the known cassava *B. tabaci* mitotypes into six SNP-based haplogroups [22]. Currently there is no knowledge of the diversity of cassava-colonizing whiteflies in the Comoros Islands.

CMBs have been reported from Comoros, an archipelago in the south-west Indian Ocean Between mainland Africa and Madagascar [23]. However, there are no country-wide reports about CMD in Comoros. Although CMD is the second most important biotic constraint to cassava, it has received less attention in Africa in recent years mainly owing to the growing CBSD threat. Nevertheless, CMD is the most important biotic constraint to cassava production in West Africa and other parts where CBSD has not been reported.

CBSD was first reported to affect cassava in the 1930s at a research station in north-eastern Tanzania [4], and remained confined to coastal East Africa for over 70 years before it started to spread at mid-altitude areas of East and Central Africa. Although until about two decades ago CMD was the only major viral disease of cassava, outbreaks of CBSD started to spread throughout East and Central Africa during the early years of the 21st century [24,25]. A series of new disease spread reports in previously unaffected areas followed [26,27,28] exacerbating concerns about food security in the Great Lakes region, where losses caused by the severe CMD outbreaks had not yet been recovered. Ever since, CBSD has progressed further into the interior of East and Central Africa [24]. CBSD causes yield losses of up to 100% in susceptible cassava varieties [29].

Recent reports have revealed the occurrence of CBSD in the Comoros Islands [30]. Similar to CMD, there is very little information about CBSD spread in Comoros. Owing to the importance of these two viral diseases, the current study conducted a detailed analysis of CBSD and CMD spread and the distribution of *B. tabaci* in Comoros and proposed strategic management approaches to ensure sustainable cassava production.

## 2. Materials and Methods

### 2.1. Survey Sites and Dates

Field surveys were conducted throughout the Comoros Islands of Mwali, Ngazidja, and Ndzwani in July 2019 where cassava fields were inspected for the presence of foliar viral disease symptoms and infestation by *B. tabaci*. Sites were selected at intervals of about three kilometers between fields along motorable roads, enabling the assessment of 66 sites across the three islands.

### 2.2. Foliar Viral Disease Symptoms and Vector Abundance Assessment

For each visited field, virus disease symptoms and abundance of *B. tabaci* were recorded from 30 cassava plants sampled at regular intervals along two diagonal axes (X pattern) through the field [31]. The predominant variety was recorded and selected for the parameters assessment. CMD foliar symptoms were scored on a scale of 1–5 where 1 = no symptoms and 5 = very severe symptoms [32]. Whitefly-borne and cutting-borne CMD symptoms were distinguished using the method of Sseruwagi et al. [31]. CBSD foliar symptoms were recorded using a scale of 1–5 where 1 = asymptomatic, 2 = mild severity, and 5 is the most severe symptoms [33]. *B. tabaci* abundance was recorded by counting adult whiteflies from the top five leaves of the tallest shoot for each of the 30 visited plants [34]. In addition, approximately 50 whiteflies were randomly aspirated from several plants in each field and preserved in 95% ethanol for subsequent genetic identification. 

### 2.3. CBSIs and CMBs Detection by (RT) PCR Testing

Wherever symptoms of virus disease were present, four CBSD and four CMD symptomatic plants were deliberately sampled at near regular intervals along the first of the two diagonal axes while two non-symptomatic plants were sampled whenever encountered at the beginning and at the end of the diagonal, making up a total of ten leaf samples per field. Wherever there were no virus symptoms in a field for either or both of CBSD and CMD, ten leaf samples were picked at regular intervals, five along each diagonal. This sample collection method aimed at maximizing the chances of selecting virus-infected plants for molecular characterization while providing a chance for detecting viruses in non-symptomatic plants. Samples were preserved by pressing between two to three layers of blank newsprint sheets in a herbaria, clearly separating the ones for CBSD and CMD testing. In this way, the leaves were left to dry and maintained at ambient temperature and moisture-free until required for further processing. 

Total nucleic acid (DNA and RNA) was extracted using a standard cetyltrimethyl ammonium bromide (CTAB) method [35] and nucleic acids were re-suspended in nuclease-free PCR-grade water. CBSIs were detected using CBSV- or UCBSV-specific TaqMan assays [36,37] using an AriaMx Real-Time PCR System (Agilent technologies, Santa Clara, CA 95051 United States). CMBs were detected using a standard multiplex PCR method for simultaneous detection of ACMV and EACMV [38]. Amplicons were separated in 1% agarose gel 1× Tris-acetate-Ethylenediaminetetraacetic acid (TAE) (Life Technologies, Grand Island, NY, USA) stained with 0.3x GelRed nucleic acid stain (Biotium, Fremont, CA, United States) and visualized under UV light. Gel images were recorded using a G-Box: Chemi XR5 Gel Documentation System (Syngene, Cambridge, UK). 

### 2.4. High Throughput Sequencing for CBSIs and CMBs

Fifty-five out of 264 samples that had typical symptoms of CBSD and 208 out of 237 with CMD symptoms did not produce positive RT-PCR or PCR tests with the methods described above. For CMBs, other methods not shown here were also employed producing similar results. A subset of these symptomatic samples that provided negative PCR results, 10 showing typical CBSD symptoms and 10 showing typical CMD symptoms, were randomly selected for next-generation sequencing. These were used to prepare the total RNA. Total RNA was extracted (using the CTAB protocol described by Maruthi et al. [35]) from dried leaf samples. The total RNA obtained was ethanol-precipitated (according to requirements established by sequencing agent: Fasteris) and transported to Fasteris, Switzerland. At Fasteris, samples were subjected to quality control and libraries for each of the 20 samples were individually prepared using a Qiagen Kit for small RNA (sRNA). Libraries were sequenced using the Illumina NextSeq500 platform.

### 2.5. Virus Detection

Small RNA sequences obtained from NextSeq500 were input into the virus detection software (VirusDetect for Windows (VDW) version 0.93; [39]) with the appropriate parameter settings for automatic library cleaning. The minimum sequence length after cleaning was set at 15 nucleotides. VDW automatically mapped input small RNA sequences to the selected plant virus database: vrl_Plants_239_U95 uploaded to VDW from GenBank. Generated contigs were then automatically BLAST-aligned (NCBI) and aligned virus sequences/genomes were generated in an output file.

### 2.6. Viral Genome Assembly

Sequence reads or contigs generated with the VDW were uploaded into CLC Genomics WorkBench (Version 21.0.3; Qiagen Aarhus, Aarhus, Denmark) with parameter settings: mapping tool selected, Burrows-Wheeler Aligner (BWA) [40]. Assembled sequences were aligned to publicly available sequences using the native alignment tool in CLC Genomics WorkBench and were manually inspected and edited for errors. Additionally, primer and probe binding sites for the routine CBSIs real-time RT-PCR [36] were inspected for mismatches using randomly selected CBSV (MK103392, Fn434437, Gu563327, HG965221, and KR911738) and UCBSV (MK103391, MK103392, KR911721, NC_014791, and MG387656) isolates available in GeneBank and aligned with the newly sequenced isolates. 

### 2.7. Genetic Identification of B. tabaci and Other Whiteflies

The whiteflies collected in this study were identified through *COI* sequencing, while those identified as cassava *B. tabaci* were further designated using KASP genotyping. Although *COI* is ineffective at distinguishing cassava *B. tabaci*, it is used because: (i) *COI* allows for comparison of findings to earlier datasets that used only *COI* for cassava *B. tabaci*, (ii) *COI* allows for separation of cassava and non-cassava *B. tabaci* as KASP is currently only applicable to cassava *B. tabaci*, (iii) *COI* allows for comparison with *B. tabaci* from other parts of the world. DNA was extracted from 168 single adult whiteflies. A partial fragment of mitochondrial DNA Cytochrome Oxidase I (*COI*) was amplified using one set of primers, 2195-Bt-F (5′-TGRTTTTTTGGTCATCCRGAAGT-3′) and C012-Bt-sh2-R (5′-TTTACTGCACTTTCTGCC-3′) [41]. Samples that failed to amplify with this first set of primers were tested using the universal primers LCO (5′-GCTCAACAAATCATAAAGATATTGG-3′) and HCO (5′-TAAACTTCAGGGTGACCAAAAAATCA-3′) [42]. The PCR reaction contained 1× QuickLoad Master Mix (New England Biolabs, UK), 1 mM MgCl_2_, 0.24 µM of each primer, 2 µL DNA, and sterile distilled water to achieve the desired reaction volume of 25 µL. PCR was carried out under the following conditions: 95 °C for 5 min for initial denaturation of template DNA, followed by 35 cycles of (94 °C for 40 s, 56 °C for 30 s for annealing, and 72 °C for 90 s for extension), with a final extension at 72 °C for 10 min. The PCR products were run on a 1% agarose gel in 1× TAE buffer stained with GelRed. DNA bands were visualized while using a Gel Doc XR+ Gel Documentation System and only samples with intact bands of the expected size (867 bp) were selected for sequencing. PCR products were sent to Macrogen Inc. (Rockville, Maryland, United States) for purification and direct sequencing. DNA sequences were manually edited using Ridom Trace Edit v1.1.0 (Ridom GmbH., Würzburg, Germany). The sequences were assembled into contigs using CLC Main Workbench 7.0.2 (QIAGEN, Aarhus, Denmark). Multiple alignment of edited sequences was performed using ClustalW in Molecular Evolutionary Genetics Analysis software (MEGA version 7.0.26) [43] and the sequences were trimmed to 669 nucleotides. Construction of a maximum likelihood phylogenetic tree was performed using MEGA with 1000 bootstrap replicates. Sequences were blasted using GenBank’s (NCBI) Blastn and selected reference sequences with 99% to 100% identity to our *COI* sequences were included in the phylogenetic tree for comparison with previously published haplotypes.

The 46 cassava *B. tabaci* samples used to generate the *COI* phylogenetic tree were tested using the KASP diagnostic with a set of six primers (BTS99-319, BTS22-762, BTS141, BTS55-473, BTS613, and BTS46203) [22]. Conventional primers were used to generate PCR products of genome portions containing target SNPs and the PCR products were then used as DNA template in KASP genotyping [22]. The KASP reaction mixture (10 µL) contained 5 µL 2× KASP master mix, 0.14 µL KASP primer assay mix and 5 µL DNA template (1 µL of PCR product/DNA extract + 4 µL of sterile water). KASP genotyping was performed in a Stratagene MX 3000P qPCR system (Agilent Technologies, Santa Clara, California, United States). The following cycling conditions were used: Stage1: 30 °C 60 s (pre-read); Stage 2: 94 °C for 15 min hot-start *Taq* activation (1 cycle); Stage3: 94 °C for 20 s, 61 °C (61 °C decreasing 0.6 °C per cycle to achieve a final annealing/extension temperature of 55 °C) for 60 s (10 cycles); Stage 4: 94 °C for 20 s, 55 °C for 60 s (29 cycles); Stage 5: 94 °C for 20 s, 57 °C for 60 s (3 cycles); Stage 6: 37 °C for 60 s (1 cycle, cooling) followed by an end-point fluorescent read. These conditions were used for four primers (BTS99-319, BTS22-762, BTS55-473, and BTS141), while Stage 3: 94 °C for 20 s, 68 °C (68 °C decreasing 0.6 °C per cycle to achieve a final annealing/extension temperature of 62 °C) was used for two primers, BTS613 and BTS46-203. The quality of genotyping cluster plots was visually assessed and only samples in distinct clusters with respective positive controls were considered for manual SNP calling using the MxPro software incorporated into the Stratagene MX 3000P unit. 

## 3. Results

### 3.1. Distribution of Cassava Varieties and Disease Symptoms

#### 3.1.1. Distribution of Cassava Varieties

Eleven varieties were encountered during the field survey (Table 1). Variety Mdja was the most frequently encountered appearing in 32 of the 66 sites visited. During the survey, crops were at growing stages of 1.5 to 12 months after planting (MAP) with mean age at 6.1 MAP.

#### 3.1.2. Distribution of CBSD and CMD Symptoms

Foliar symptoms of CBSD and CMD were widely distributed throughout the Comoros islands of Mwali, Ngazidja, and Ndzwani. No spatial disease distribution pattern was obvious across all islands, as both diseases were present in all parts of all islands. Overall CBSD (mean severity score 2.86) and CMD (mean severity score 3.00) symptoms were moderate to severe (Figure 1, Table 1) and no statistically significant differences were observed between islands. 

Nevertheless, a wide range of leaf and stem symptoms were observed throughout the visited sites (Figure 2). CBSD and CMD were prevalent throughout Comoros: CBSD occurring in 83.3% and CMD in 95.5% of fields. CBSD was present in all fields surveyed in Mwali (Table 1). Mean CMD incidence was 31.6%. There was no statistically significant difference in total CMD incidences between islands and there were also no statistically significant differences for the whitefly- and cutting-borne CMD incidences (Table 2). Overall mean CBSD leaf incidence was 42.0% and there were no statistically significant differences between islands although Mwali and Ngazidja had relatively higher leaf CBSD incidence than Ndzwani (Table 2). There were generally low incidences of CBSD stem necrosis, ranging from ~3 to 10% (Table 2). Whereas no statistically significant associations were observed between crop age and viral disease symptoms, altitude (using only data for the most frequent variety, Mdja) showed a statistically significant negative relationship with foliar incidences of CBSD (r = −0.41, *p* = 0.026), total CMD (r = −0.52, *p* = 0.003), cutting-borne CMD (r = −0.38, *p* = 0.037), and whitefly-borne CMD (r = −0.53, *p* = 0.0025). All varieties expressed symptoms of either or both of CBSD and CMD.

### 3.2. CBSIs and CMBs Detection by (RT) PCR Testing and Next-Generation Sequencing

CBSV- and UCBSV-specific TaqMan assay results showed that only CBSV was detected across the Comoros islands of Mwali, Ngazidja, and Ndzwani. Overall, 53.3% of all 330 (221 symptomatic and 109 non-symptomatic) tested samples were positive for CBSV. Neither CBSV nor UCBSV was detected in samples with typical CBSD symptoms from seven of the 55 sites where CBSD symptoms were observed (12.7%) equal to 24.9% of all symptomatic leaf samples (plants) tested. Interestingly, no CBSIs were detected in samples from 11 sites that were recorded as non-symptomatic (i.e., no single plant observed with CBSD) during field assessment. However, CBSV was detected from a few samples collected from non-symptomatic plants in sites that also had symptomatic plants. There was a significantly higher (Kruskal-Wallis chi-squared = 14.76, df = 2, *p* < 0.001) frequency of CBSV detection in Mwali (78.8%) compared with Ngazidja (56.6%) and Ndzwani (27.0%). Mwali also had the highest congruency between symptom expression and virus-positive test detection of all the three islands visited (97.1%) whereas Ngazidja had the largest number of negative symptomatic samples followed by Ndzwani (Table 3).

CMBs were detected in only 29 of the 330 tested samples, all with EACMV (data not shown).

### 3.3. Virus Detection and Discovery by VirusDetect

Viruses detected with contig coverages, >50% of corresponding reference genomes, were selected and determined as true viruses based on suggestions put forward by Zheng et al. [39]. For CBSIs, both CBSV and UCBSV were detected whereas only EACMV isolates were detected for CMBs. While only one virus type was detected for some of the sequenced samples, multiple infections of either CBSIs or CMBs were frequent (Appendix A). All the full-length and partial viral sequences detected for each sequenced sample are shown in Appendix A. However, two of the 20 sequenced samples did not produce quality reads and therefore no viruses were detected for those samples. Only assembled full-length genomes for both CBSV, UCBSV, and EACMV were submitted to GenBank and assigned accession numbers (Table 4). It is noteworthy that no other (novel or already known) viruses/viroids were detected with the VirusDetect software on CBSD and CMD cassava affected plants assessed in this study. BLAST results of the CBSV isolate with accession number MZ362877, from this study, showed that it was 98.6% identical to a CBSV isolate from Comoros with GeneBank accession number MK103392 while that of UCBSV (MZ362878) from this study was most closely related (93.5%) to a UCBSV isolate from Comoros (GenBank accession number MK103391). Investigation of the primer and probe binding sites of the routinely used Adams et al. [36] primers and probes for diagnosis of CBSIs indicated mismatches at different positions in probes/primers binding sites with the new CBSIs (MZ362877 (CBSV) and MZ362878 (UCBSV)) and all the available complete sequences previously published on Comorian CBSV ((MK103392) and UCBSV (MK1033391 and MK103393)) isolates (Table 5). However, no mismatches were detected for the new and previously published Comorian isolates with the forward primer of CBSV while there were also no mismatches between the reverse primer and the new UCBSV (MZ362878) sequence.

### 3.4. Characterization of B. tabaci and Other Whiteflies

#### 3.4.1. Vector Abundance 

Low counts of *B. tabaci* whiteflies were recorded in Comoros throughout the entire survey area. The overall mean *B. tabaci* abundance was <2 insects per plant (Table 1). There was no notable statistical difference in whitefly abundance across the three Comoros islands visited, nor were there any statistically significant difference in *B. tabaci* distribution across altitudes.

#### 3.4.2. Genetic Diversity of Whiteflies

A total of 131 whitefly samples produced quality sequences, of which 116 were with the 2195-Bt-F/C012-Bt-sh2-R primers and 15 with the LCO/HCO primers. The *Bemisia* spp. comprised 116 samples of which 46 (40%) were cassava *B. tabaci*, 6 (5%) were non-cassava *B. tabaci,* and 64 (55%) were *Bemisia afer* (Priesner and Hosny). The 15 non-*Bemisia* samples included 12 spiralling whitefly (*Aleurodicus dispersus* Russell) and three Bondar’s nesting whitefly (*Paraleyrodes bondari* Peracchi). A phylogenetic tree (Figure 3) generated for the 52 *B. tabaci* whiteflies revealed 41 (79%) were in a mitotype that was very close to SSA1-SG2. These have 100% identity to only two samples in GenBank (KF425628) and have three nucleotide differences in the 669bp fragment compared with KM377899-SSA1-SG2 (samples from Uganda and Malawi). Five (10%) of the samples were mitotype SSA1-SG3 while the remaining 6 (11%) were Indian Ocean (IO). The *B. afer* samples were clustered into two major groups with clade 1 comprising 66% and clade 2 having 32% of the samples. The spiralling whitefly formed a single clade as did the Bondar’s nesting whitefly.

KASP genotyping of cassava *B. tabaci* (46 samples designated using mtCOI) revealed 38 (83%) samples were SSA-ESA while the remaining 8 (17%) were heterozygous, all of which were categorized as mitotype SSA1-SG2. This is the first report of KASP-typed SSA-ESA whiteflies comprising samples with a mitotype that is not SSA1-SG3. In previous studies [44,45] all samples of mitotype SSA1-SG2 were SNP designated as SSA-ECA or SSA-CA.

## 4. Discussion

Comoros, similar to other African countries, remains disadvantaged as its cassava yields are significantly below the global average. The Food and Agriculture Organization listed average cassava yield as 5.7 t/ha in Comoros for the year 2020 which is nearly four times less than the average obtained by Asian farmers in the same period of 21.9 t/ha [2]. Damage from cassava pests and diseases is thought to be a major cause of such low yields, which highlights the importance of characterizing the status of these biotic constraints, the two most important of which are anticipated to be CMD and CBSD. The study reported here attempted to characterize the epidemiology of CMD, CBSD, and the whitefly vectors transmitting the viruses that cause them.

### 4.1. Varietal Response to Cassava Viruses

All the varieties encountered were severely affected by either CBSD or CMD and in most cases by both. Typical symptoms of both diseases were observed on all varieties indicating that they are all susceptible to CBSD and CMD. The low incidence (7.1%) of severe CBSD stem symptoms and dieback recorded in the current work suggests that the encountered varieties have a degree of tolerance to CBSD which is a common feature of cassava germplasm in East Africa, and which contrasts with germplasm introduced from West Africa which often has catastrophic root damage when affected by CBSD [46]. The scope of the current study did not include assessment of roots which can be used to ascertain the level of loss caused by CBSD which frequently results in corky dry necrotic rot that renders roots of susceptible varieties unfit for human consumption and marketing. However, the low incidence of severe stem symptoms and dieback recorded in the current study suggests that varieties are partially tolerant to CBSD. The need for improved cassava varieties was emphasized in a report showing yearly increase in cassava yield which was attributed to increase in cultivated land area rather than variety performance [47]. Incidences of CMD cutting infection were greater than those for CMD whitefly infection on each of the three islands, demonstrating that CMD is an endemic disease in Comoros that has likely been present for many years. CMD has been recorded throughout mainland East Africa and Madagascar for many decades.

### 4.2. Characterization of Cassava Brown Streak Ipomoviruses and Cassava Mosaic Begomoviruses

Virus testing on samples collected during the field survey revealed wide-spread occurrence of CBSD- and CMD-associated viruses throughout the Comorian islands. Viral disease prevalence was high (greater than 80%) both for CBSD and CMD across the Comorian Archipelago. Although there is report of the presence of CBSD causing ipomoviruses (CBSIs) in Comoros [30] as well as CMD causing begomoviruses, there is no comprehensive data on the prevalence and incidences of these viral diseases published for Comoros. Isolates of CBSV and UCBSV were reported and confirmed by Sanger sequencing using PCR products obtained with primers designed by Mbanzibwa et al. [8]. In this study, however, these primers failed to produce amplicons with most of the CBSD symptomatic samples collected (data not shown). Although the real-time RT-PCR primers and probes of Adams et al. [36] provided better results, at least 24% (55/221) of samples with typical CBSD symptoms were negative with this assay. Our research team is making efforts and experiments to optimize these assays to detect more virus isolates. Re-testing of these samples with a novel assay would change the virus incidence in Comoros. Furthermore, these results highlight the importance of continuous viral disease surveillance in Comoros and updating of the diagnostics protocols to cope with viral sequence changes and strain variation. Recent data from other locations in East Africa have recorded a shift in the frequencies of species detection where CBSV or UCBSV and mixed infections were distributed unequally over the length of planting seasons [48,49]. Examination of primer/probe binding sites revealed presence of mismatches between Comorian CBSIs isolates with some of the primers and probes of Adams et al. [36] which is the routinely used method for detection and quantification of CBSIs. The presence of mismatches in probe binding sites has been implicated for false negative real-time RT-PCR studies elsewhere [50]. The Adams et al. [36] assay was designed at a time when there were no sequences available from Comoros, but which were later identified to contain isolates that were phylogenetically distant from other East African isolates [51]. Next-generation sequencing offers a suitable approach in addressing challenges of false negative results such as the results obtained in this study where isolates of both CBSIs (CBSV and UCBSV) were detected, supporting earlier findings of the presence of CBSV and UCBSV in Comoros [30,51]. The sequences of CBSV isolates from the study reported here shared the highest nucleotide identity of 98.6% with the previously identified Comorian isolate with GenBank accession number MK103392 while that of UCBSV shared the highest nucleotide identity of 93.5% with isolate MK103391, also from Comoros. The highest CBSV nucleotide identity with non-Comorian isolates was 95.3% with an isolate from Uganda (GenBank accession MW961165), whereas the highest UCBSV nucleotide identity with non-Comorian UCBSV isolates was 78.4%, which is consistent with the findings of Scussel et al. [51] who reported that the UCBSV isolates from Comoros represent a different lineage from UCBSV strains previously reported in East Africa. These results explain the failure of existing diagnostic tools to pick up UCBSV isolates from Comoros and highlight the necessity of developing new primer sets specifically designed for Comorian isolates of CBSIs.

EACMV was the only CMB detected. ACMV has not been reported previously from Comoros and we did not detect any during the current study. CMD is distributed wherever cassava is grown in Africa [52] with at least eight species widely distributed across sub-Saharan Africa [53]. Although ACMV is the most widely distributed of all CMBs [54,55], ACMV has never been reported from coastal East Africa, so it is unsurprising that it also seems to be absent from Comoros. There are, however, reports of ACMV from Mozambique [56] and Madagascar [57]. Since mixed infections of ACMV and EACMV-like viruses (of which there are many) result in more severe disease because of virus-virus synergy [58,59], any introduction of ACMV to Comoros represents a significant threat to their cassava production (and therefore food security). Globally, there are 11 distinct CMB species (International Committee on Taxonomy of Viruses (ICTV)). Of these, only two occur in South Asia while the remainder are distributed through parts of SSA. None has been reported from South America where cassava originated. ACMV is predominant in West Africa but widely distributed in sub-Saharan Africa except in the coastal areas of Kenya and Tanzania: EACMV and EACMV-like CMBs occur throughout Central, East, and Southern Africa with the exception of East African cassava mosaic Cameroon virus (EACMCV) which is the predominant EACMV-like virus occurring in West Africa and African cassava mosaic Burkina Faso virus (ACMBFV) which occurs only in West Africa [60]. Regular investigation on the distribution and epidemiology of the CMBs will help improve our understanding of the relationship between the existence of these viruses and how they affect the overall economic and food security in SSA. The fact that no new viruses/virions were detected in cassava samples studied in this work confirms the association of the detected viruses with the observed diseases.

### 4.3. Mismatch between Disease Incidence and Virus Incidence

We report in this study widespread distributions of CBSD and CMD throughout the Comoros Archipelago. However, there is no evidence for major differences in virus transmission throughout the studied area. Nevertheless, a small difference between CBSV and UCBSV transmission has been reported in a previous study [61] showing CBSV as having higher transmissibility by the vector *B. tabaci*. In this study, whitefly abundance was generally low suggesting that most of the recorded virus infections were due to infections spread through infected planting material. It is generally recognized that CMB virus presence is more closely linked to CMD symptoms than CBSI presence and CBSD symptoms. This contrasts with the results of the current study in which many plants with CMD symptoms tested negative with available PCR diagnostics. This suggests that the primers, developed against strains from other geographic locations, were not effective in detecting CMBs from Comoros. HTS confirmed that all the CMD symptomatic samples that tested negative with PCR were in fact infected by CMBs. The same was true for samples that had CBSD symptoms, but which were PCR-negative. This validates the use of symptom-based scoring applied in this study, although it is recognized that there will never be a perfect match between symptom expression and virus presence, particularly for CBSD where symptoms are often cryptic.

### 4.4. Characterization of Bemisia tabaci Whiteflies

This study reports the abundance and genetic diversity of whiteflies found on cassava in the Comoros Islands after an extensive survey. The low whitefly population observed can be attributed to sampling season which was cool and the age of the plants which were mature. A previous study shows whitefly numbers are low during the cool season which is occasioned by low temperatures and long rains, and whitefly populations decline as cassava plants mature beyond six months [51]. The clustering of the cassava-colonizing *B. tabaci* into two mitotypes, SSA1-SG2 and SSA1-SG3, that belong to the major clade of SSA1 is similar to whiteflies that have been reported in other cassava-growing regions of East and Central Africa [19,44,62,63,64]. However, SSA1-SG2 was the dominant mitotype accounting for 79% of the cassava *B. tabaci*. This finding contrasts with the pattern in most cassava-growing countries in East and Central Africa in which SSA1-SG1 is the dominant mitotype [19,44]. It also differs from previous studies considering *B. tabaci* from coastal East Africa which only reported mitotype SSA1-SG3 [19,44]. This is also the first report of mitotype SSA1-SG2 outside continental Africa as so far it has only been reported from Central African Republic, Democratic Republic of Congo, Kenya, Tanzania, Uganda, and Zambia [19,62,64,65].

KASP genotyping of the 46 cassava *B. tabaci* once again proved that mtCOI is ineffective at distinguishing the cassava *B. tabaci* whiteflies as previously reported [44]. All five samples in mitotype SSA1-SG3 were designated as haplogroup SSA-ESA which was consistent with previous findings [44,45]. Of the 41 samples in mitotype SSA1-SG2, 33 were designated in haplogroup SSA-ESA while the remaining 8 were heterozygous. This is the first time SSA1-SG2 samples have been grouped by the more robust KASP diagnostics as SSA-ESA; all previous samples from continental Africa in the SSA1-SG2 mitotype were designated as either SSA-ECA (large majority) or SSA-CA (minority) [44,45]. This result is unsurprising, however, as so far, all cassava *B. tabaci* from south coastal Tanzania, Mozambique and Madagascar regions, which are in proximity with the Comoros Islands, have been designated in the SNP-based haplogroup SSA-ESA [44,45].

The non-cassava *B. tabaci* Indian Ocean mitotype was found in samples collected from cassava. This mitotype has been occasionally reported from samples collected from cassava plants [44,63,64,66,67,68], but it is not known to reproduce on and colonize cassava [66,68,69]. It is therefore concluded that non-cassava *B. tabaci* mitotypes occurring on cassava are not colonizers but transient visitors. *Bemisia afer* is known to colonize cassava, and previous studies report the existence of two clades in samples collected from cassava fields in Kenya, Malawi, Tanzania, and Uganda [64,70]. The high proportion of *B. afer* can also be attributed to the age of the cassava plants which were mostly older than 6MAP, as this species is known to colonize older plants as opposed to *B. tabaci* which prefers young plants less than six months of age. Additionally, the high proportion of *B. afer* can also be attributed to the prevailing weather condition which was the cool season during the survey in Comoros in July 2019. *B. afer* is known to prefer cool temperatures [71]. *B. afer* is unlikely to be of any major economic significance in Comoros, however, as it is not known to transmit CMBs or CBSIs and it has never been reported to occur at high abundance levels or cause any kind of damage to cassava. 

Spiralling whitefly and Bondar’s nesting whitefly are polyphagous invasive species that have recently spread globally probably due to increased trade in plant products and change in climate [72]. Although in this study they occurred in small numbers, they have been reported to colonize and co-exist on cassava and are known to cause low to moderate damage in this crop [72]. The spiralling whitefly has been reported from various countries across Africa [64,73,74,75,76,77]. Bondar’s nesting whitefly has been reported to colonize cassava in Uganda, Nigeria, and Tanzania [74,77,78] and was found in whitefly samples from cassava in coastal Kenya [64].

Whitefly diagnostics are critical to determine species diversity, distribution, and change in composition over time, and expansion of geographical and host range. This knowledge is important in evaluating the potential impact of these whiteflies on cassava production through virus transmission and or physical damage in the Comoros Islands. Developing and deploying effective control strategies especially biological agents such as parasitoids/predators and entomopathogenic fungi is dependent on knowing the pest species [72]. Management with synthetic insecticides also requires knowledge of species due to increased rates of insecticide resistance development especially among the MEAM1 and MED mitotypes of *B. tabaci* [79]. 

## 5. Conclusions

We report a comprehensive surveillance study of cassava viral diseases and the associated whitefly vector *B. tabaci* in the Comoros Archipelago. A wide diversity of cassava varieties is cultivated throughout the Comoros islands of Ngazidja, Mwali, and Ndzwani. However, all of these are susceptible to the viruses that cause CBSD and CMD. CBSIs and CMBs were widely distributed throughout the islands. Although the occurrence of CBSIs reported here matched previous studies, and CBSD symptoms were observed in all the three islands, relatively high levels of false negative results during molecular testing using existing PCR primers showed that there is a need to develop diagnostic assays that will achieve more reliable cassava virus detection.

*Bemisia tabaci* abundance was low throughout the surveyed areas. Since this is the vector for the viruses that cause both CBSD and CMD, it would be prudent to conduct similar surveys at different times of the year to record the patterns of abundance change over time and identify periods when levels of vector-borne virus infection are greatest. Although we report a high prevalence of both cassava virus diseases in Comoros, high levels of virus resistance have been developed through conventional breeding programs elsewhere [80,81]. Both diseases can be effectively controlled through the introduction of sources of resistance coupled with local breeding work. Finally, it is recommended to establish a strong cassava seed system in Comoros to facilitate the delivery of healthy planting material of disease-resistant varieties as they become available. Additionally, this study characterized four whitefly species occurring on cassava. The dominant cassava *B. tabaci*, based on SNP genotyping, is SSA-ESA, which has been reported in the countries in proximity with Comoros. Although low abundances were recorded at the time of the study in July, populations are likely to be greater during hotter, wetter periods of the year. Long-term solutions to whitefly-transmitted viruses affecting cassava in Comoros, as elsewhere, will depend on combining host plant virus resistance with strong phytosanitary measures and effective whitefly management.

## Figures and Tables

**Figure 1 viruses-14-02165-f001:**
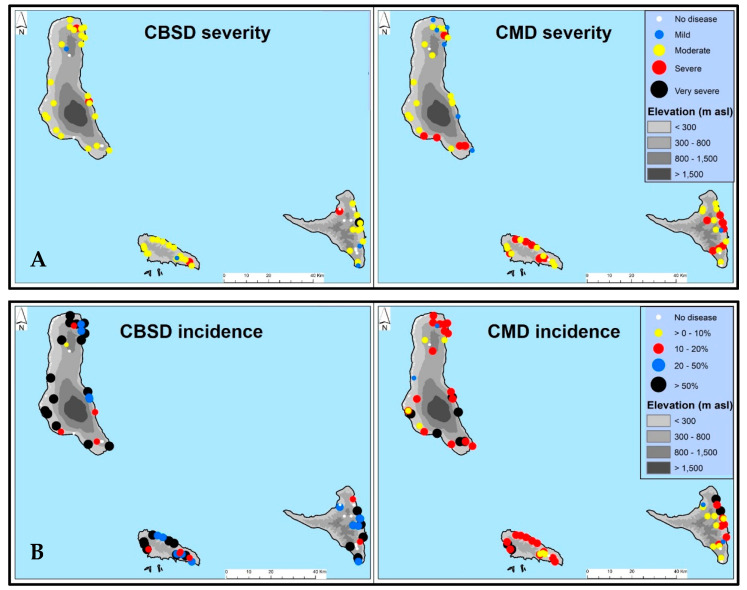
Map of Comoros showing the distribution of cassava brown streak disease and cassava mosaic disease in Comoros, July 2019. CBSD: cassava brown streak disease; CMD: cassava mosaic disease. (**A**) CBSD and CMD severities in Comoros, July 2019. Disease severity scores were determined according to Cours, (1951) [32] and Gondwe et al. (2003) [33]. The colored bullets represent disease scores where: white = score 1 (no symptoms), blue = average score 2 (2 < 3; mild symptoms), yellow = average score 3 (moderate symptom), red = average score 4 (3.1 < 4; severe symptom), and black = average score 5 (scores 4.1 ≤ 5; very severe symptom). (**B**) CBSD and CMD incidences in Comoros, July 2019.

**Figure 2 viruses-14-02165-f002:**
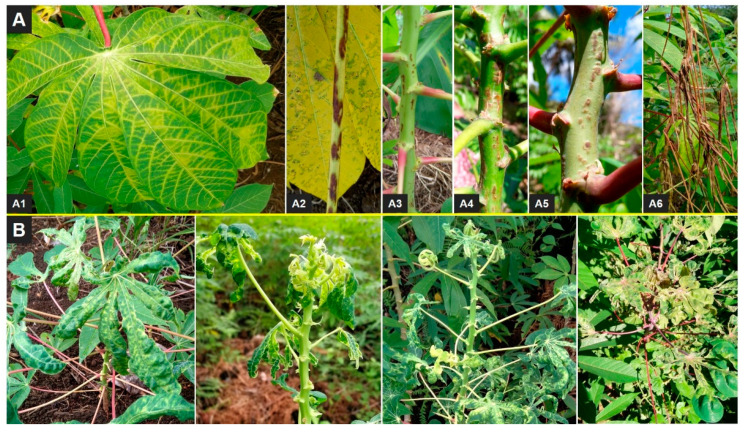
Foliar symptoms of cassava brown streak disease (CBSD) and cassava mosaic disease (CMD) observed in Comoros, July, 2019. Panel (**A**): Different foliar symptoms of CBSD observed on different varieties in Comoros: (**A1**), feathery necrotic lesions on leaf; (**A2**), necrotic lesions on leaf petiole; (**A3**–**A5**), different patterns of necrotic lesions observed on the green parts of shoots of different varieties; (**A6**), shoot of cassava showing severe necrotic lesions and dieback. Panel (**B**): CMD symptom patterns observed on different cassava varieties in Comoros.

**Figure 3 viruses-14-02165-f003:**
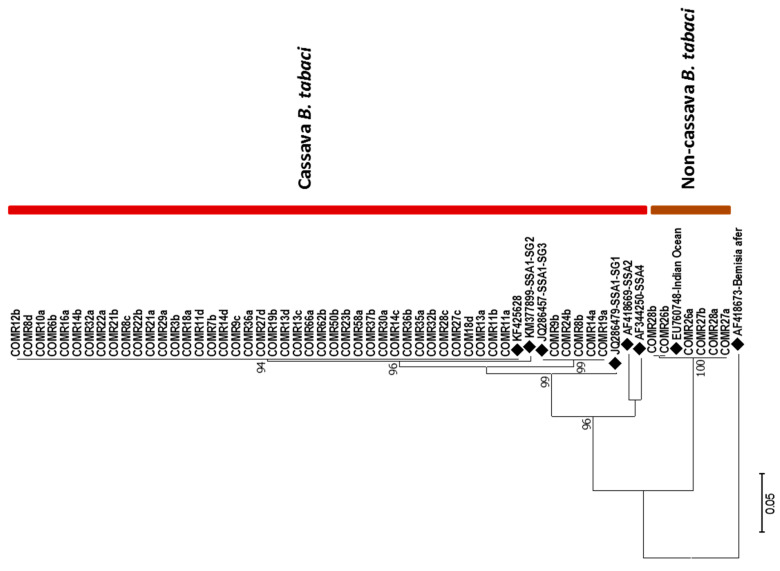
Maximum likelihood phylogenetic tree constructed for COI sequences obtained from *Bemisia tabaci* collected from Comoros Islands in July 2019. Reference sequences from GenBank (♦) are included for comparison. *Bemisia afer*, AF418673, was included as an outgroup. The numbers at the nodes represent bootstrap values.

**Table 1 viruses-14-02165-t001:** Incidence, prevalence, and severity of cassava virus diseases and abundance of the whitefly *Bemisia tabaci* for cultivars surveyed in islands of Comoros, July 2019.

Island	Variety	Sites	^$^ *Bemisia tabaci*	^#^ Leaf CBSD Severity	^&^ Leaf CBSD Incidence	^#^ CMD Severity	^&^ CMD Incidence	^^^ CBSD Prevalence	^^^ CMD Prevalence
Ndzwani	Chihawati	2	0.92	2.33	10.0	3.90	25.0	65.0	95.0
	Java	2	0.17	3.04	41.7	2.71	20.0		
	Mdja	8	2.05	2.79	43.3	3.18	35.8		
	Meladi	2	3.02	*	0.0	2.80	8.3		
	Mkoudu	1	0.57	*	0.0	2.56	30.0		
	Unknown	1	1.77	*	0.0	3.00	3.3		
	Wachididri	4	1.26	2.97	35.0	3.05	21.7		
Mwali	Chihawati	2	3.85	2.77	66.7	3.25	26.7	100.0	94.1
	Mdja	6	2.97	2.99	51.1	3.01	41.7		
	Mdjomani	1	4.87	2.60	16.7	3.00	53.3		
	Meladi	3	1.23	2.72	24.4	2.96	23.3		
	Mweou	4	0.29	2.91	71.7	3.01	25.0		
	Unknown	1	0.47	2.60	16.7	3.83	20.0		
Ngazidja	Java	3	1.82	2.85	24.4	3.18	54.4	86.2	96.6
	Mdja	18	1.37	2.86	56.3	2.73	30.6		
	Mdjema	2	3.97	*	0.0	3.53	56.7		
	Mdjomani	3	1.89	2.98	47.8	2.95	21.1		
	Mkoudu	1	2.13	2.92	40.0	3.07	50.0		
	Nkatsa	1	1.37	2.76	56.7	2.91	36.7		
	Unknown	1	0.77	2.71	23.3	2.86	46.7		
Total/Mean		66	1.75	2.86	42.0	3.00	31.6	83.3	95.5

***** No foliar symptoms observed, CBSD: cassava brown streak disease; CMD: cassava mosaic disease; **^$^**
*Bemisia tabaci*: whitefly abundance which is the mean number of adult *B. tabaci* insects counted from five fully open top leaves of each assessed plant in a given site (determined according to Sseruwagi et al. [31]): **^#^** Leaf CBSD/CMD severity: average severity score (classes 2–5) calculated for the 30 assessed plants per site; **^&^** Leaf CBSD/CMD incidence: percentage of plants showing visible foliar CBSD or CMD symptoms in a visited site. Disease severity scores were determined according to Cours [32] for CMD and Gondwe et al. [33] for CBSD. **^^^** CBSD/CMD: the proportion of CBSD/CMD affected sites in the given Island in Comoros.

**Table 2 viruses-14-02165-t002:** Comparison of cassava brown streak disease, cassava mosaic disease symptoms, and *Bemisia tabaci* abundance across three islands in Comoros, July 2019.

Island	Sites	*Bemisia tabaci*	CMD Sev.	CBSD Sev.	CMD Wf	CMD Cut	CMD Total	fCBSD Inc.	stCBSD
Mwali	17	2.10	3.08	2.85	10.2	22.0	32.2	49.0	10.4
Ngazidja	29	1.66	2.88	2.86	9.2	26.1	35.3	46.6	7.8
Ndzwani	20	1.60	3.12	2.84	5.0	20.7	25.7	29.5	3.2
Average/Total	66	1.75	3.00	2.86	8.2	23.4	31.6	42.0	7.07

CBSD: cassava brown streak disease; CMD: cassava mosaic disease; Sev: severity; Inc.: incidence; CMDWf: CMD incidence of whitefly-borne infections; CMDCut: CMD incidence of cutting-borne (infected planting material) infections; CMDTotal: overall CMD incidence (CMDWf + CMDCut); fCBSD Inc.: CBSD leaf incidence; stCBSD: CBSD stem incidence.

**Table 3 viruses-14-02165-t003:** Real-time RT-qPCR and PCR testing results for cassava brown streak ipomoviruses and cassava mosaic begomoviruses in cassava leaves collected from field plants in Comoros, July 2019.

**Real-Time RT-PCR Testing for Symptomatic Samples**					
**Island**	**Number of samples collected**	**CBSV positive**	**UCBSV positive**	**CBSIs negative**	**%Symptomatic positive**
Mwali	69	63	0	6	91.3
Ngazidja	99	73	0	27	73.7
Ndzwani	53	27	0	26	50.9
Total	221	163	0	59	
**Real-time RT-PCR testing for asymptomatic samples**					
**Island**	**Number of samples collected**	**CBSV positive**	**UCBSV positive**	**CBSIs negative**	**%Asymptomatic positive**
Mwali	16	4	0	12	25.0
Ngazidja	46	9	0	36	19.6
Ndzwani	47	0	0	47	0.0
Total	109	13	0	95	
**PCR testing for CMD symptomatic samples**					
**Island**	**Number of samples collected**	**ACMV positive**	**EACMV positive**	**CMBs negative**	**%Symptomatic positive**
Mwali	61	0	6	55	9.8
Ngazidja	95	0	14	81	14.7
Ndzwani	66	0	8	58	12.1
Total	222	0	28	194	
**PCR testing for CMD asymptomatic samples**					
**Island**	**Number of samples collected**	**ACMV positive**	**EACMV positive**	**CMBs negative**	**%Asymptomatic positive**
Mwali	24	0	0	24	0.0
Ngazidja	50	0	1	49	2.0
Ndzwani	34	0	0	34	0.0
Total	108	0	1	107	

**Table 4 viruses-14-02165-t004:** Full genome sequences of cassava brown streak ipomoviruses and cassava mosaic geminiviruses isolated from different locations in Comoros, July 2019.

Sample ID	BankIt Submission ID	GenBank Accession Number	Sites ID	Virus
AQHM1C	BankIt2465160	MZ362877	NgB1.1	CBSV
AQHM2U	BankIt2465160	MZ362878	NgB19.5	UCBSV
AQHM1EA4-B	BankIt2470742	MZ494476	NgB1.1	EACMV
AQHM20EA2-B	BankIt2470742	MZ494477	NgM16.1	EACMV
AQHM23EA4-B	BankIt2470742	MZ494478	AnM19.4	EACMV
AQHM25EA1-B	BankIt2470742	MZ494479	AnM13.5	EACMV
AQHM26EA1-B	BankIt2470742	MZ494480	MoM3.3	EACMV
AQHM29EA3-B	BankIt2470742	MZ494481	MoM12.2	EACMV
AQHM1EA1-A	BankIt2473359	MZ494482	NgB1.1	EACMV
AQHM20EA1-A	BankIt2473359	MZ494483	NgM16.1	EACMV
AQHM23EA3-A	BankIt2473359	MZ494484	AnM19.4	EACMV
AQHM25EA2-A	BankIt2473359	MZ494485	Anm13.5	EACMV
AQHM26EA2-A	BankIt2473359	MZ494486	MoM3.3	EACMV
AQHM27EA1-A	BankIt2473359	MZ494487	MoM3.4	EACMV

CBSV: cassava brown streak virus; UCBSV: Ugandan cassava brown streak virus; EACMV: East African cassava mosaic virus.

**Table 5 viruses-14-02165-t005:** Mismatches in primer/probe binding sites on sequences of cassava brown streak ipomovirus isolates from Comoros.

Accession Number	* Primer or Probe	^ Mismatch	^$^ Mismatch Position	Number of Mismatches
MZ362877	CBSV probe	A/T	14	1
MK103392	CBSV probe	A/G	2	1
MK103392	CBSV reverse	T/C and G/A	5 and 15	2
MK103391 and MZ362877	UCBSV forward	A/G	2	1
MK103391 and MZ362877	UCBSV probe	A/T and T/A	6 and 21	2
MK103391 and MZ362877	UCBSV reverse	A/G	4	1

* Primers and probes used from Adams et al. [36]. ^ The numerator represents a nucleotide in the target sequence while denominator represents a nucleotide in the primer or probe used, 5′-3′ shows the direction of sequences from five prime to three prime, **^$^** Positions were assigned on the primer or probe counting the first nucleotide at the 5′ end as number 1.

## Data Availability

All relevant data are presented in this article.

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
