# Peer review of "Epidemiological Analysis of Cassava Mosaic and Brown Streak Diseases, and *Bemisia tabaci* in the Comoros Islands"

_viruses, 2022, doi:10.3390/v14102165_

Round 1
Reviewer 1 Report
The manuscript entitled “Epidemiological analysis of cassava mosaic and brown streak diseases, and Bemisia tabaci in Comoros” by Rufini Shirima and colleagues reports the viral status in cassava in Comoros islands, the spread and incidence of two major diseases, CMD and CBSD as well as the genetic characterization of collected whiteflies. The manuscript is generally concise and designed well. Nevertheless, there are a few points that need improvements before final publication (I quote them below); therefore, my recommendation is acceptance with revision.
L3. Add “islands” in the title, next to Comoros
L43. I would expect to read in this section, after the two phrases on CMB, a couple more about CBSD. You mention in L59-60, but perhaps in this section you should add the importance and spread of CBSD in Africa, and then you continue with the status and the current data in Comoros islands.
L45. In the introduction, you must refer to the virus species that are associated with CMD and CBSD (like you did in the abstract)
L47. Are these mitotypes morphologically similar or identical?
L113. So, 20 samples were analyzed individually with HTS? Or you pooled some samples? In Table 5, 14 Sample IDs are referred. In these 14 samples, only single virus infections were detected?
L130. Figure 1 could be omitted from the manuscript, as the data on HTS results are also presented in the manuscript. If the authors still want to incorporate this figure, the most appropriate way is to include the dataset from all analyzed samples as a supplementary table/figure.
L198. I believe this Table (Table 1) needs further clarifications in the legend (since they should be self-explanatory). What do the numbers in each column represent (this mean severity score how is it calculated)?
L207. Statistically significant differences? If not, how/why are they (or not) significant? On what criteria? Revise these terms in the entire MS.
L234. Data from Table 2 can be incorporated in Table 1, as they both include information on the disease incidence.
L244. Perhaps Table 3 should move to supplementary?
L266. Table 4 should include all the results from RT-PCR or TaqMan RT-PCR tests on viruses, CBSV, UCBSV, ACMV, EACMV (including results on L270-271). Also, which ipomoviruses do you mean? Isn’t it only CBSV (with TaqMan)? Moreover, the way that the columns with the results are presented, it is hard to follow the status on symptomatic and asymptomatic plants. Maybe you should split for each island the virus status (not the disease status) on symptomatic and asymptomatic plants, and this should be done for all the four viruses you tested (even if the results were negative). This would create a very informative Table and help the readers to understand your findings and keep up with the MS flow.
L251/L269: Delete headlines 3.2.1 and 3.2.2
L304: notable statistical difference?
L327. Add in the figure legend which sequence was used as an outgroup and what the numbers at the node represent.
L365: replace ‘causing’ with ‘associated’
L376: here, you present novel results (with reference 50) that are not included neither in materials and methods nor in the Results section and they are not part of another study. I am not sure if this reference ‘50’ can be included in the MS as it shows a manuscript under preparation and not an accepted manuscript (with DOI). I understand that you tried to improve the primer set so as to detect more UCBSV isolates but unless you incorporate these info (materials and methods and results) in the MS, you cannot mention these results at all. You should commend that efforts and experiments are made by your research team to optimize these assays so as to detect more virus isolates. And that re-testing of these samples with a novel/improved assay would change the virus incidence.
L363-423. It is important to mention (if this is the case) that no other (novel or already known) viruses/viroids were detected with the software on CSBD and CMD cassava plants (in the results section) and this indicates a stronger correlation of the detected viruses with the observed diseases (in the discussion part)
L441. From the ‘once again’, I understand that you expected these findings. So, this is another case where mtCOI seems to be inefficient for distinguishing these whiteflies. So, my question is why did you use mtCOI in the first place? For comparison of the techniques? If yes, you should add this purpose in the materials and methods section and/or in the results section (with the appropriate headlines)
L453. This mitotype has been occasionally reported…
A general comment: in the entire MS, the authors choose to present the results on the grounds of viral diseases (CMD and CBSD) and not of associated viruses (CBSV, UCBSV, ACMV, EACMV). I rarely came up to the names of viruses in the MS. I believe it is important to record the disease status but more important is to document the viruses that are present in these islands (and their association with CMD and CBSD), as any changes in viral species status (in single or multiple infections) would have serious implications in the observed diseases in the near future. Moreover, each virus species has (most likely) a different capacity/efficiency of whitefly transmission, which affects directly the epidemiology part. Therefore, my recommendation is to re-shape these parts (not all the parts) referring to diseases based on these grounds.
Reviewer 2 Report
The study is interesting with sufficient details provided to the reader. However, there are minor improvements suggested to enhance the readability. They are as follows:
Table 1: what is meant by “CBSD incidence” and “CMD incidence”? Is it percentage?
Line 142: Please explain CBSI. Explained in line 368, but the same should be done at this place.
Line 274: Change the structure of the sentence. The reference should be cited after a statement.
Line 331-346: This part is not necessary. It is best to focus only on the results obtained.
Line 415: SSA needs to be explained.
Reviewer 3 Report
This manuscript studied the distribution, severity, and the viruses infecting cassava crop. The reuslts obtained in this study are useful to know the virus speices infecting cassava crop in Comoros. However, some results are somehow confusing. E.g. The author collected samples with CMD symptoms, however, most of them were not positive using RT-PCR. Athough, the small RNA sequencing results revealed the existing of EACMV in the mixed samples, it is still not clear the incidence of CMD in the samples. Furthermore, it is also make no sense descriping as CMD severity. It is not accurate to descript the distribution just based on symptom. Further PCR detection on the collected samples are needed to be sure the distribution of different viruses.
other comments, the name for CBSV and UCBSV, e.g. "cassava brown streak ipomoviruses (CBSIs) and cassava mosaic begomoviruses (CMBs) ", need to be revised.
Figure 1, should be presented as Table
Title 3.1.2, the title need to be revised, we could not study the distribution of symptoms.
